# A New COVID-19 Detection Method Based on CSK/QAM Visible Light Communication and Machine Learning

**DOI:** 10.3390/s23031533

**Published:** 2023-01-30

**Authors:** Ismael Soto, Raul Zamorano-Illanes, Raimundo Becerra, Pablo Palacios Játiva, Cesar A. Azurdia-Meza, Wilson Alavia, Verónica García, Muhammad Ijaz, David Zabala-Blanco

**Affiliations:** 1CIMTT, Department of Electrical Engineering, Universidad de Santiago de Chile, Santiago 9170124, Chile; 2Department of Electrical Engineering, Universidad de Chile, Santiago 8370451, Chile; 3Escuela de Informática y Telecomunicaciones, Universidad Diego Portales, Santiago 8370190, Chile; 4Departamento en Ciencia y Tecnología de los Alimentos, de la Universidad de Santiago de Chile, Santiago 9170124, Chile; 5Manchester Metropolitan University, Manchester M1 5GD, UK; 6Department of Computer Science and Industry, Universidad Católica del Maule, Talca 3480112, Chile

**Keywords:** COVID-19, CSK, QAM, VLC, BER

## Abstract

This article proposes a novel method for detecting coronavirus disease 2019 (COVID-19) in an underground channel using visible light communication (VLC) and machine learning (ML). We present mathematical models of COVID-19 Deoxyribose Nucleic Acid (DNA) gene transfer in regular square constellations using a CSK/QAM-based VLC system. ML algorithms are used to classify the bands present in each electrophoresis sample according to whether the band corresponds to a positive, negative, or ladder sample during the search for the optimal model. Complexity studies reveal that the square constellation $N=2^{2^{i}}\times 2^{2^{i}},\left(i=3\right)$ yields a greater profit. Performance studies indicate that, for BER = $10^{-3}$, there are gains of −10 [dB], −3 [dB], 3 [dB], and 5 [dB] for $N=2^{2^{i}}\times 2^{2^{i}},\left(i=0,1,2,3\right)$, respectively. Based on a total of 630 COVID-19 samples, the best model is shown to be XGBoots, which demonstrated an accuracy of 96.03%, greater than that of the other models, and a recall of 99% for positive values.

## 1. Introduction

While the severe acute respiratory syndrome coronavirus 2 (SARS-CoV-2) is still undergoing new mutations, it is currently risky to declare that the virus is no longer a problem. It is unknown whether the current vaccines prevent severe symptoms, hospitalisation or death. Some research on new COVID-19 variants indicates that the virus is spreading faster than in the past [1] and has effects of maternal SARS-CoV-2 infection on pregnant women, foetuses, and newborns [2]. A study on the presence of COVID-19 and its association with respiratory syncytial virus was conducted during the winter of 2020–2021 in Europe and North America [3] to determine whether SARS-CoV-2 mutates similarly globally or whether it mutates differently in specific populations [4]. Newly emerging variants of SARS-CoV-2 continue to pose a significant threat to global public health by causing COVID-19 epidemics [5]. The SARS-CoV-2 pandemic has highlighted the need for routine monitoring of infections in high-density indoor areas, such as hospitals and underground environments, with the strictest monitoring required for dust particles in pollution-absorbing tunnels and metro stations.

The primary method for preventing transmission is social distancing, for which measurement mechanisms have been developed that establish areas of risk based on the number of people in a given geographical area [6]; however, the collection of medical samples does not allow the health system to maintain constant monitoring, because it requires patients to visit a medical centre or have medical personnel visit their home. In addition, only variables such as temperature [7] and physical conditions can be monitored continuously [8].

This strategy was tested in many areas during the SARS-CoV-2 pandemic and was validated by a number of authors [9,10]. Another benefit of pathogen identification through sewage is the ability to monitor both endemic and Waterborne Datasets (WBDs) [9,11,12,13]. The latter are collections of microorganisms primarily related to diarrhoea disorders that are transferred by water or food irrigated in polluted water and generate large-scale outbreaks. A national pathogen monitoring system utilising Optical Wireless Communications (OWC) technologies will provide a significant contribution and is an innovative technique for detecting these pathogens. Individual testing and traceability systems are one way to achieve this objective. However, the cost of the other detection methods and the limited representativeness of the gathered data prevent the creation of appropriate models for this objective. It reduces the number of samples required for analysis, is representative of the population whose waste is channelled into the sample, is independent of sanitary conditions due to the low availability of tests, enables the observation of under-represented or asymptomatic diseases, and is less expensive than other methods [14]. The use of mathematical methods on images in conjunction with ML techniques yields results that aid in less subjective decisions being made, allowing later validation of the diagnosis [15]. Among other things, the impacts of the epidemic on public health, culture, the environment, and the economy [16] have provided motivation for the use of technologies such as Artificial Intelligence (AI), ML [17], robotics [18], big data [19], and the IoT [16]. In order to measure vital signs, many of the mechanisms employed for this purpose include equipment that must come into contact with the body of an individual [20,21] by using specimens from medical facilities [22,23] or at home through the use of robots [18].

In relation to this paper, our team has published original models of underground channels [24,25,26,27,28]. Additionally, before the pandemic, we produced papers on DNA [29,30,31] and work on MIMO [32,33,34]. Pathogens will always be around humans, so it is crucial to conduct channel studies that allow us to transfer information under any circumstance in a secure and fast manner.

This manuscripts makes the following contributions to this area of research: It provides a mathematical model for mapping DNA genes using a CSK/QAM scheme transmitted by Frequency Shift Keying (FSK) over a MIMO VLC-based underground channel and an ML-based technique for identifying COVID-19.

In addition to this introduction, there are four more sections in this document. In Section 2, the current state of knowledge is presented. Section 3 provides the methodology. Section 4 is a discussion of the results, while Section 5 contains the conclusion.

## 2. State-of-the-Art Techniques

This section presents the state-of-the-art techniques used for the application of VLC in underground channels. This is followed by the presentation of a model of a channel based on colour shift keying with quadrature Amplitude modulation (CSK/QAM) mapping, the use of the Galois Field Mapping/Galois Fields Demapping interface between the human side of the machine and the signal processing communication channel, and finally, AI-based procedures to replace human processing. Note that COVID-19 and pathogens in general are most likely to spread rapidly in indoor settings, such as hospitals or industrial settings like mine tunnels. As evidenced by events of the last three years, this can result in a decrease in a country’s GDP and crippling of its economy.

### 2.1. Work Related to the VLC Channel

As mentioned in the previous paragraph, the fact that gene information can be colour-coded makes it necessary to discuss alternatives for the implementation of the underground channel. The above model can be used in interior settings, hospitals and underground tunnels. For this reason, studies on underground channels, scattering distribution patterns, and FSK in an underground channel are presented. Using experimental tests and mathematical simulations, it has been found that FSK is the best method for reaching longer distances, because the energy is concentrated in a single frequency tone in a mining tunnel.

#### 2.1.1. Work Related to Underground Channels

Earlier efforts modelled the VLC channel in underground mining environments using the same Lambertian channel model as an indoor VLC channel. This meant that dust scattering, reflections on uneven walls, light obstruction, also known as shadowing, and the relative tilt and rotation of LEDs and PDs were disregarded. Furthermore, light has a dual nature and, depending on the quantum interpretation of the observer, can be modelled as either a particle or a wave [35]. The majority of current research on VLC communication in underground channels focuses on on–off keying (OOK) modulation, which is similar to amplitude modulation. In terms of phase, frequency modulation includes both coherent and noncoherent signals. Coherent FSK signals are those whose phase remains constant over time. Inconsistency also exists when the phase changes or varies over time. The continuous use of pneumatic hammers to excavate rocks in underground tunnels generates a substantial quantity of airborne dust that is detrimental to any channel. Prior generation processes have always sought to include [36] VLC or hybrid communication. To model the scattering effect and incorporate it into the overall model of the underground mining visible light communication channel (UM-VLC), a robust mathematical infrastructure is required for UM-VLC [37,38]. Consequently, the effect of dust particles in the air is disregarded during testing in nonscattering indoor environments, such as offices, homes and hospitals; consequently, such models cannot function in underground tunnels.

#### 2.1.2. Work on Scattering Distribution Patterns

When the dust particle size is very small, the proposed dispersion distribution models can be implemented in any type of multipath wireless communication system. However, as the dust particle size increases and the dust concentration causes small holes in the sensors, the waves revert to particle behaviour and scatter more. Previous studies modelled different arrival schemes over time and various communication scenarios [39,40,41]. A dust disc around the optical receiver was modelled as a uniform distribution within a 2D disc region [42,43]. In [40], a statistical analysis was conducted in a hemispheric area around a base station. Using a geometric model of a mobile transmitter channel, the signal’s arrival time and direction were analysed. A Gaussian scatter distribution model was presented. Due to the model’s spatial–temporal properties, in terms of the arrival angle and arrival time [41], it will eventually be possible to apply it to multipath wireless communication systems. Tennskoon [44] proposed a three-dimensional (3D) stochastic geometry model with a Gaussian distribution centred on an arbitrary point within a sphere.

#### 2.1.3. Work Related to FSK in Underground Channels

As previously stated, the majority of the literature on VLC in general has used OOK and laboratory level tests at very short distances with white light. On the ground, this results in large, power-hungry drivers. Therefore, it might be interesting to include the frequencies or wavelengths of the chosen colours [45]. There are few FSK applications for VLC which, by definition, has a longer range and lower power consumption. In [46], an advertising panel that uses FSK to communicate with a cell phone application is discussed. Salmento [47] described a lab-scale VLC system comprised of a single-stage buck-boost power factor correction converter operating in discontinuous drive mode with dimming capability. Dahri [48] described a system for vehicle-to-vehicle communication using FSK. The only studies on FSK modulation applied to underground mining are presented in [49,50,51], which all involved testing in a mine tunnel.

The models discussed previously are extremely rigid and linearly conceptualised. Quantum objects, on the other hand, do not need to have their properties defined; a beam of light can arrive at the photodetector not only in a straight line in a coherent manner but also via other angles in an incoherent manner [52]. Dust, for instance, can be modelled as large particles colliding with waves, which cause collisions that deflect the waves, but photon jets arrive at the photodetector because they are aligned. If these become stuck in the detector and clogged by dust, leaving a few holes that convert the photon jets back into waves, all of the previously discussed models become invalid. According to Aharonov [53], infrastructure maintenance and cleanliness appear to be more important than simple model validation.

#### 2.1.4. Model of a Channel Based on CSK/QAM Mapping

In the context of visible light communication, the use of the colour shift keying modulation scheme has allowed the application of different techniques in relation to the optimisation of the colour space which, according to the IEEE 802.15.7 standard, presents nine valid schemes for the combination of colours with 4, 8 and 16 Color Shift Keying (CSK) constellations. However, performance should be improved by concatenating other modulation or coding methodologies. To improve spectrally efficient transmissions in VLC systems using CSK communication, coding and mapping techniques, such as bit-interleaved coded modulation with iterative demapping and decoding [54], are applied. These work best for high-speed VLC applications. Machine learning is also applied to find the most optimal combination of coding and modulation technologies. Its performance is based on the mapping of symbol permutations through points in an optimized CSK constellation, This offers benefits in terms of diversity, resistance to channel degradation monochrome, and increased security. This type of method can be combined with MIMO technologies [54] to evaluate systems through Monte Carlo simulations. For the generation of a multiuser channel, the optimisation of channel resources and the technical type of spatial division used are essential. Separation by means of CSK modulation techniques to maximise the minimum Euclidean distance between different points of a constellation or multiuser joint constellation [55] is used for this purpose. Techniques such as multiplexing of the symbols used in wireless channels where each 7-bit 128 QAM symbol is multiplexed by a complex value signal to form a 32 QAM with an additional gain of 40% is done to compensate for problems related to chromatic dispersion and non-Kerr linearity [56]. The use of constellation probability shaping is a high-order modulation format optimisation technology that optimises the probability distribution of each signal constellation point to improve the generalised mutual information and increase the transmission capacity of QAM modulation [57].

### 2.2. Galois Field Mapping/Galois Fields Demapping

In [58], the author describes the problems faced and efforts to eradicate the COVID-19 pandemic. In order to achieve this objective, documentation is produced to examine the signatures of genomes using chaotic studies. First, alternative representations of the SAR-COV-2 DNA sequences, such as colour-coded images, indicator matrices, DNA walks, and chaotic games were created.

In [59], the detection of cancer using images is proposed. Cells are constantly exposed to numerous mutagens that produce diverse types of DNA lesions. Eukaryotic cells have evolved to contain a vast array of DNA repair mechanisms that are capable of detecting and repairing these lesions, thereby preventing genomic instability. Based on their functions, repair proteins are recruited to lesions sequentially.

In [60], the helitrons, eukaryotic transposable elements transposed by the rolling-circle mechanism, are defined. These have been identified in numerous species with highly variable copy numbers and, in some cases, they comprise a significant portion of the genome. Using images of the constituent helitron features and a pretrained neural network as a classifier, classification was conducted using the *k*-means features corresponding to genomic sequences, and this method was compared with the Support Vector Machine (SVM) and Random Forest methods.

A few studies have employed Galois fields to numerically represent DNA [29,31]. Representation through Galois Fields GF(p) is based on gel electrophoresis, a standard method for separating double-stranded DNA (dsDNA) fragments of different sizes previously obtained by the Polymerisation Chain Reaction. When interpreting the electrophoresis of GF(p) and its extension $GF(p^{n})$, the standard notation [61,62] is utilised. Pathogen-causing SARS-CoV-2 DNA is used for COVID-19 detection because it contains four distinct genes: Adenine (A), Cytosine (C), Guanine (G), and Thymine (T). By using four non-binary symbols, the four states $2^{2^{i}},i=1$ can be represented by natural numbers such as $\left[0,1,2,3\right]$ or by colours such as $\left[red,green,blue,yellow\right]$ [63,64].

In [59], the author proposed the detection of cancer using images. Cells are constantly exposed to a variety of mutagens that generate various types of DNA lesions. In order to prevent genomic instability, eukaryotic cells have evolved a vast array of DNA repair mechanisms capable of detecting and repairing these lesions. According to their function, repair proteins are sequentially recruited to lesions.

Reference [60] described how helitrons, eukaryotic transposable elements (tes) transposed by the rolling-circle mechanism, have been identified in numerous species with highly variable copy numbers and, in some cases, constitute a significant portion of the genome. Using images of the constituent helitron features and a pretrained neural network as a classifier, classification using the *k*-means features that correspond to genomic sequences was conducted, and a comparison to the SVM and Random Forest methods was made.

Many studies have linked ML to the diagnosis of COVID-19 using lung X-rays [23,65,66]. Deep neural networks (DNN) were used to process images in [67], and statistical methods were used in conjunction with heuristic filtering to identify somatic mutations in tumour samples.

### 2.3. AI-Based Procedures to Replace Human Processing

In the manual detection of SARS-CoV-2, a machine–human couple interpreted the gel electrophoresis results following a two-step end point Reverse Transcriptase PCR (RT-PCR). In this method, the $N_{1}$ and $N_{2}$ gene targets are followed to detect SARS-CoV-2, and Ribonuclease P (RNase
*P*) is used for Ribonucleic acid (RNA) extraction. A dataset of 242 gel images obtained in that study was utilised in this work [68].

Using a histogram database, [69] contributed to the formation of a stratification system with three severity levels (moderate, severe and mild) that defines infection in various slides from a COVID-19 patient. The authors of [70] argue that the use of deep learning in medical imaging is an emerging technology for the diagnosis of a variety of diseases, such as pneumonia, lung cancer, brain stroke, and breast cancer. Before constructing a predictive model, machine learning and conventional data mining techniques perform the time-consuming feature extraction process. A convolutional neural network (CNN) was constructed using 1920 Chest X-rays (CxR) from healthy individuals and COVID-19 infected patients as training data. Using the clinical results of the 300-CxR validation dataset, the performance of the developed CNN was assessed further.

## 3. System Diagram

Figure 1 is a system diagram that illustrates the phases of model searching and operation of the classification model. Assuming the diagram can be folded along the dashed line, the first three boxes at the emitter and the last three boxes at the receiver represent human activity that could be automated. The organic samples consist of chromosomes with four genes represented by four symbols, Adenine (A), Cytosine (C), Guanine (G), and Thymine (T), which are passed through thermocycles or undergo amplification of the deoxyribonucleic acid (DNA), separation and delivery to the next block. The ”(+)”, ”(−)”, and “Ladder” states of the receiver’s reverse function, designated by the listing symbol, reveal the genes that contain COVID-19.

At the entrance of the second block, the resulting DNA samples are loaded into the second block, called “Electrophoresis”, which allows the generation of images of gels. The inverse block called “Artificial Intelligence Classification” represents the best model for classification.

In the third block, called “Computer Vision Processing”, the image from the previous block is filtered and delivered to the next block for numerical representation. On the receiving side, the reverse process is called “Computer vision interpretation”, where a numerical input is converted into an image for interpretation.

The remaining issues are associated with the communication process. Galois Field Mapping/Galois Fields Demapping is a function that converts images to polynomials in the emissor and polynomials to images in the receiver. In the fifth block, called “CSK/QAM modulation”, each of the numbers is calculated as the centroid of the CSK modulation and mapped into a QAM constellation, and these signals are sent over the MIMO channel. In the receiver, the block called “CSK/QAM demodulation” takes the QAM signals and converts them into numbers which are delivered to the Galois Fields Demapping block.

Figure 1 depicts a massive array of LEDs and photodetectors inside a box that represents the VLC/FSK MIMO channel. The segments connecting the antennas represent electromagnetic waves propagating in air molecules as photon jets or sine waves with an amplitude, frequency, and phase.

### 3.1. Line-of-Sight (LoS) Link

To obtain $h_{\mathrm{LoS}}\left(t\right)$, the most fundamental VLC link is considered with a single light source (LS), which can be monochromatic or multichromatic, and a single PD in an indoor free-space environment. When considering the LS, a point source from the perspective of the PD, the optical received power $P_{R,\mathrm{opt}}$ can be expressed as [71]:(1)$$P_{R,\mathrm{opt}}=P_{T,\mathrm{opt}}G_{\mathrm{conc}\;}G_{\mathrm{filter}\;}f\left(\theta ,\theta_{1/2}\right)\frac{A_{R,\;\mathrm{eff}\;}}{\pi r^{2}}\;\mathrm{for}\;d\gg \lambda \;\mathrm{and}\;A_{R}\gg \lambda^{2},$$
where $P_{T,\mathrm{opt}}$ is the optical transmission power, $G_{\mathrm{conc}\;}\geq 1$ is the optical concentration gain, $G_{\mathrm{filter}\;}\leq 1$ is the optical filtering loss, *r* is the distance between the LS and the PD, $\theta_{1/2}$ is the half-power angle of the light beam, $A_{R}$ is the aperture area of the PD, and $A_{R,\mathrm{eff}\;}$ is the effective aperture area of the PD such that [71]:(2)$$A_{R,\mathrm{eff}}=A_{R}\cos\phi ,$$

Note that the condition d≫λ stems from the point source assumption, while the condition $A_{R}\gg \lambda^{2}$ implies that the optical power detection process at the PD is deterministic. Note that when ϕ exceeds the field of vision (FOV) of the PD, $\phi_{\mathrm{POV}}$. Then, $P_{R,\mathrm{opt}}=0$ [71].

### 3.2. Non-Line-of-Sight (NLoS) Link

Multipath channels cause stochastic and time-varying signal distortion in Radio Frequency (RF) communications, causing microwave channels to be modelled as random. The multipath channel in the VLC, on the other hand, is deterministic because $A_{R}\gg \lambda^{2}$. In other words, the PD captures the optical signal over an area that is millions of times larger than a square wavelength. The indoor VLC channel is time-invariant as long as the objects in the room are fixed. Nonetheless, multipath propagation can cause intersymbol interference in VLC systems at high data rates, according to Hoeher [71]. To obtain $h_{\mathrm{NLoS}}\left(t\right)$, it is easiest to begin with a single reflector. This reflector serves as a virtual light source (VLS). Because most reflections are diffuse, the angle of irradiance $\theta_{2}$ is not always the same as the angle of incidence $\phi_{1}$. Furthermore, Lambertian reflections are commonly used. By using $r_{1}$ to represent the distance between LS and VLS and $r_{2}$ to represent the distance between VLS and PD, $G_{\mathrm{conc}}G_{\mathrm{filter}\;}=1$, and Equation (Equation 3) can be extended to [71]:(3)$$P_{R,\mathrm{opt}}=P_{T,\mathrm{opt}}f\left(\theta_{1},\theta_{1/2}\right)\frac{A_{\mathrm{ref},\mathrm{eff}}}{\pi r_{1}^{2}}\cdot \rho \left(\lambda \right)f\left(\theta_{2},60^{\circ }\right)\frac{A_{R,\mathrm{eff}}}{\pi r_{2}^{2}}$$

### 3.3. Transmitter: LED

The semiconductor light sources known as light-emitting diodes (LED) emit light when current flows through them. This is conceivable because of the electroluminescence phenomenon, in which the forward current causes semiconductor electrons to rejoin electron holes and release energy as photons. The energy required for electrons to traverse the band gap of the semiconductor determines the wavelength of the emitted light. Phosphor LEDs and Red Green Blue (RGB) LEDs are the two most commonly used types of LED to generate white light: (i) by using a blue LED with yellow phosphor, white light is produced and (ii) by using the RGB-based LED, which does not use phosphorus and is sustainable, high speeds can be reached in domestic environments [72].

### 3.4. Receiver

Photodiodes are photoelectric transducers because they convert signals of optical power to electrical impulses. The photodiode will generate an output current that is proportional to the incident optical power R(λ) at a specific wavelength Φ(λ). It is measured in amperes per watt (A/W), as shown by Equation (Equation 4). The materials and structure of a photodiode determine its response curve.
(4)$$R\left(\lambda \right)=\frac{i_{r}}{\Phi (\lambda )},$$

### 3.5. The DC Gain of the Channel Model

When an optical input and optical output are considered, the Direct Current (DC) gain of the channel model is given by
(5)$$H_{\mathrm{ref},}\;\mathrm{opt}=H_{\mathrm{LoS}}+H_{\mathrm{NLoS},(1)}+H_{\mathrm{scatter}}\;,$$
where $H_{\mathrm{LoS}}$ is the LoS component, $H_{\mathrm{NLoS},(1)}$ is the single-hop NLoS component, and $H_{\mathrm{scatter}}$ is the NLoS contribution of light scattering off dust particles. Due to the insignificance of the following bounces in terms of the received power and time dispersion, only one hop is evaluated [25,37]. In Appendix A, we provide descriptions of reference model equations, which describe the most applicable reference models utilised for the UM-VLC Single input single output channel (UM-VLC SISO) [24,25]. In [38], it is assumed that particles are spread through a two-dimensional disc with irregular walls, nondeterministic diffuse reflections, shadows, and a scattering component.

## 4. Methodology

From right to left, Figure 1 shows the methodology, which comprises the MIMO channel, CSK/QAM Modulation and the DNA strand picture as a polynomial and vice-versa.

### 4.1. Model of a Channel Based on CSK/QAM Mapping

Due to its low cost and simplicity, intensity modulation with direct detection (IM/DD) is used in the majority of VLC systems. The transmitters in this kind of system are LEDs, and the instantaneous optical power $\Phi_{e}\left(t\right)$ is modulated in proportion to the driving electrical current $i_{t}\left(t\right)$, which is modulated in accordance with the data to be broadcast.

The optical power signal travels down the channel and eventually reaches the receiver’s surface, which is often a photodiode, also known as a photodetector (PD). In the photodiode, the received optical power causes a proportionate photocurrent, $i_{r}\left(t\right)$. In this study, the Bit Error Rate (BER) was calculated using the Minimum Mean Square Error (MMSE) estimator for a variety of MIMO arrays [73]. In order to send the DNA samples that are in a sewer, an uplink process must be created. This assumes that there are $N_{t}$ light sources capable of transmitting the signal and a receiving device with $N_{r}$ Photo detector (PDs):(6)$$y=\left(\begin{array}{ccc} h_{1,1} & \ldots & h_{1,N_{t}}\\ \; & \ldots & \\ h_{n_{r},1} & \ldots & h_{n_{r},N_{t}} \end{array}\right)x+n\;,$$
where y is the $N_{r}\times 1$ vector representing the signal received at each PD, x is the transmitted signal vector of size $N_{t}\times 1$, $\begin{array}{ccc} h_{i,j}(i=1,\ldots ,N_{r} & and & j=1,\ldots ,N_{t}) \end{array}$ are the channel gain of the link between the *j*-th transmitter and the *i*-th PD, and n is the $N_{r}\times 1$ vector representing the noise at each PD, including all possible noise, which can be expressed as Equation (Equation 5).

Furthermore, the mapping of an M-CSK constellation to an M-QAM constellation is proposed. Figure 2 represents an xyY diagram of the Commission Internationale de l’Éclairage (CIE) colour space from 1931. This was the first colour space based on experimental results of human colour perception. A colour space is a multidimensional collection of all colours that a certain colour model can generate. Historically, the IEEE 802.15.7 standard defines some guidelines for designing M-CSK constellations and directly applying them for modulation [74].

Figure 3 illustrates the fundamental principle of mapping for the chromosomal length $N^{DNA}$. Depending on the length of the DNA, $N^{DNA}=2^{2^{i}}$ points in the M-CSK constellation are selected and mapped to the M-QAM constellation, which is depicted on the right side of Figure 3 as clouds of points, with a total of $2^{2^{i}}$ points i=0,1,2,3… shown.

Figure 4 represents 4-CSK constellation mapping into 4-QAM. Although points in the CIE 1931 xyY space can be assigned in any arbitrary order, the four 4-CSK stars correspond to the four 4-QAM stars. Due to their proximity, it is more difficult to pass separator hyperplanes to detect the four points of the 4-CSK constellation. On the right is a 4-QAM constellation with more evenly spaced stars, which makes it simpler to deliver separator planes to them.

The centre of the colour bands can be expressed in CIE 1931 xyY space coordinates as $s_{R},s_{G},s_{B}\in R^{2}$, which is known as the centre of band symbols. All colours that can be reproduced by LEDs via additive mixing form a triangle in CIE xyY space with the vertices $s_{R},s_{G}$ and $s_{B}$.

The gamut of the system is the set of all colours that may be reproduced by the three LEDs and is mathematically defined as the convex combination of the centre of band symbols in the CIE xyY space, as shown in
(7)$$G=\left\{a_{1}s_{R}+a_{2}s_{C}+a_{3}s_{B}\in R^{2}\right\};\;a_{1}+a_{2}+a_{3}=1\wedge \forall a_{1},a_{2},a_{3}\geq 0$$
where G denotes the system’s gamut.

The explicit CIE xyY coordinates of the symbol $s_{k}$ are denoted as $s_{k}={\left[x_{k},y_{k}\right]}^{T}$. The symbol $s_{k}$ can be expressed as a radiant flux vector $\Phi ={\left[\Phi_{R},\Phi_{G},\Phi_{B}\right]}^{T}\in R_{\geq 0}^{3}$, where $\Phi_{R},\Phi_{G},\Phi_{B}$ represent the radiant fluxes to be emitted by the red, green and blue LEDs respectively. The radiant flux vector Φ can be obtained by solving the system of Equation (Equation 8).
(8)$$\begin{array}{cc} x_{k} & =\Phi_{R}x_{R}+\Phi_{G}x_{G}+\Phi_{B}x_{B}\\ y_{k} & =\Phi_{R}y_{R}+\Phi_{G}y_{G}+\Phi_{B}y_{B}\\ 1 & =\Phi_{R}+\Phi_{G}+\Phi_{B} \end{array}$$

The centroid is determined by multiplying the four points $\left(x_{k},y_{k}\right)\left(x_{R},y_{R}\right)(x_{G},y_{G})$$\left(x_{B},y_{B}\right)$ by their components and Φ, which enables the transmission of white light between the points. For separator planes, such as using a support vector machine, it is simpler to map these four points in a square constellation.

Using the cursors from the CSK, these points are mapped once more into a square regular M-QAM constellation. Data are processed by the M-QAM modulator, which then maps them into a plane with an in-phase and quadrature component.
(9)$$s=a+jb\;\mathrm{where}\;a,b\in \{\pm 1,\pm 3,\ldots ,\pm \left(\left\lceil \sqrt{M}\right\rceil -1\right)\}$$

Suppose we have *k* cursors according to Equation (Equation 41), which indicates that all $2^{2^{i}},\left(i=3\right)$ M-QAM elements of constellation have been addressed. The fastest method would be to undergo transmission in a single cycle, but this is unnecessary because transmittion as a matrix $2^{2^{i}}\times 2^{2^{i}},\left(i=1\right)$ could occur, in which case the tables would be smaller according to G.

Given a set of LEDs with $2^{2^{i}}$ colours with the same characteristics, except for having different Semiconductor Photo Detectors (SPDs), given by $\Phi_{1}\left(\lambda \right)$, $\Phi_{2}\left(\lambda \right)$⋯$\Phi_{2^{2^{i}}}\left(\lambda \right)$ respectively, spatially grouped so that their positions in space can be approximated from a sufficiently large distance and DPs, each with a spectral response R(λ) and surface area $A_{R}$ and filtered by an optical filter of one of the $2^{2^{i}}$ colours with spectral gains of $G_{1}\left(\lambda \right)$, $G_{2}\left(\lambda \right)$ …$G_{2^{2^{i}}}\left(\lambda \right)$, respectively, and spatially grouped in such a way that spatial positions can be approximated from a sufficient distance, if the distance *d* between the LEDs and PDs is large enough that a single emitter and receiver position is a reasonable approximation, then the gain of the UM-VLC DC electro-optical channel from the *i*-th LED to the *j*-th PD with $i,j=1,2,3,\cdots 2^{2^{i}}$ can be given by:(10)$$H_{\mathrm{el}}\left[i,j\right]=H_{\mathrm{LoS}}^{\mathrm{el}}\left[i,j\right]+H_{\mathrm{NLoS}}^{\mathrm{el}}\left[i,j\right]\;,$$
where $H_{\mathrm{LoS}}^{\mathrm{el}}\left[i,j\right]$ is the DC gain of the LoS link, which can be expressed as:(11)$$\begin{array}{cc} H_{\mathrm{LoS}}^{\mathrm{el}}\left[i,j\right] & =\frac{1}{P_{T,\;i}}\int_{}\;R\left(\lambda \right)A_{0,\;j}\left(\lambda \right)\Phi_{i}\left(\lambda \right)\;d\lambda \end{array}$$(12)$$\begin{array}{cc} \; & \;\;\;\;\;\;\;\;\;\;\;=C\frac{\left(m+1\right)A_{R}}{2\pi d^{2}P_{T,\;i}}\;\Omega_{\mathrm{LoS}}\Psi_{\mathrm{LoS}}\int_{}\;R\left(\lambda \right)G_{j}\left(\lambda \right)\zeta_{0}\left(\lambda \right)\Phi_{i}\left(\lambda \right)\;d\lambda \;, \end{array}$$
where $P_{T,\;i}=\int \Phi_{i}\left(\lambda \right)\;d\lambda$. Similarly, $H_{\mathrm{NLoS}}^{\mathrm{el}}\left[i,j\right]$ is the DC gain of the single-hop NLoS link, which can be expressed as:(13)HNLoSel[i,j]=∑w=1WHNLoS,wel[i,j](14)=1PT,i∑w=1W∫R(λ)Aw,j(λ)Φi(λ)dλ(15)=(m+1)AR2πPT,i∑w=1WCwArefl,wd1,w2d2,w2ΩNLoS,i(w)ΨNLoS(w)(16)∫R(λ)ρw(λ)Gw,j(λ)ζw(λ)Φi(λ)dλ(17)=(m+1)AR2πPT,i∑w=1WCwArefl,wd1,w2d2,w2ΩNLoS,i(w)ΨNLoS(w)×∑λ=1λ=22iCwR(λ)ρw(λ)Gw,j(λ)ζw(λ)Φi(λ),

It should be noted that the integral is discretised and that $2^{2^{i}}$ points are taken into account. If it is considered that the optical filters do not depend on the angle of incidence $\theta_{w}$, the definition can be simplified to $G_{w,\;j}\left(\lambda \right)\equiv G_{j}\left(\lambda \right)$. $\Omega_{\mathrm{NLoS},\;i}^{\left(w\right)}$ depends on the *i*-th LED, because light bouncing on the *w*-th reflector will have different angles. This is to account for the irregularity of the underground tunnel walls, where a small shift in the LS can have a large effect on the angle of reflection off the wall. Because the LEDs are slightly separated from one another, light from different LEDs will bounce off the walls independently. The channel impulse response for the UM-VLC channel between the *i*-th LED and the *j*-th PD is then given by [71]:(18)$$h\left[i,j\right]\left(t\right)=H_{\mathrm{LoS}}^{\mathrm{el}}\left[i,j\right]\cdot \delta \left(t-\frac{d}{c}\right)+\sum_{w-1}^{W}H_{\mathrm{NL}o\;S,w}^{\mathrm{cl}}\left[i,j\right]\cdot \delta \left(t-\frac{d_{1,w}+d_{2,w}}{c}\right),$$

Given an optical power signal $x\left(t\right)={\left[x_{1}\left(t\right),x_{2}\left(t\right),\ldots x_{2^{2^{i}}}\left(t\right)\right]}^{T}$ (*W*) as input, where $x_{i}\left(t\right)$ is the optical power signal emitted by the i−th LED, for $i=1,2,\ldots ,2^{2^{i}}$, then the received photocurrent signal at the j−th PD, $y_{j}\left(t\right)$ (*A*) can be given by:(19)$$y_{j}\left(t\right)=\sum_{i=1}^{2^{2^{i}}}h\left[i,j\right]\left(t\right)*x_{i}\left(t\right)+n_{j}\left(t\right),$$
where $n_{j}\left(t\right)$ is the noise at the *j*-th PD with noise variance of $\sigma_{j}^{2}$.

### 4.2. Galois Field Mapping/Galois Fields Demapping

This section describe how to convert a dsDNA image into a polynomial and vice versa as well as how to colour-code strands or dsDNA fragments using M-CSK/M-QAM modulation. A chromosome contains a single long molecule of DNA, only part of which corresponds to an individual gene. We developed a simple DNA-based model to represent the fields GF(p) and $GF\left(p^{n}\right),n>1$. It is based on the differential migration of dsDNA fragments of different sizes in gel electrophoresis, which is a standard technique for dsDNA fragments of different sizes that have previously been obtained by PCR. Here, the size of a dsDNA fragment corresponds to the number of base pairs [bp] that are contained in the fragment.

Each element r∈GF(p) is represented by a dsDNA fragment whose size is unique to the element *r*. Therefore, only *p* dsDNA fragments are necessary to represent all elements of GF(p). Table 1 shows this representation using dsDNA fragments of different sizes, where the smallest size $S_{0}$ is composed of one or more genes and the largest is $S_{p-1}$.

Gel electrophoresis is used to visualize the DNA molecular representation of a nonzero element $\alpha^{k}\in GF\left(p^{n}\right)$, which represents the coefficients of the polynomial expression given for Equation (Equation 20), as shown in Table 2.

The dsDNA fragments for each coefficient $a_{i}\in GF\left(p\right),i=0,1,\ldots ,n-1$ are loaded into different slots of the agarose gel matrix. The slots and their respective columns are numbered n−1,n−2,⋯,2,1,0 according to the order of powers $\alpha^{n-1},\ldots ,\alpha^{2},\alpha ,1$ from left to right. Then, an electric field is applied to force the molecules to migrate through the gel and be separated by size.
(20)$$\alpha^{k}=2\alpha^{n-1}+\left(p-1\right)\alpha^{n-2}+\cdots +\alpha^{2}+\left(p-1\right)\alpha +\left(p-1\right).$$

For this purpose, chains of size $S_{2}$ were loaded into slot n−1, chains of size $S_{p-1}$ were loaded into slot n−2, and from slot n−3 to slot 3, chains of size $S_{0}$ were loaded. Finally, chains of size $S_{1}$ were loaded into slot 2, and chains of size $S_{p-1}$ were loaded into slots 1 and 0. Thus, our model defines a unique DNA-based representation for each element of $GF\left(p^{n}\right)$.

We should note that $\alpha^{0}=1$, and the null element $0\in GF\left(p^{n}\right)$ does not have a representation as a power of α. Hence, the field $GF\left(p^{n}\right)$ has $p^{n}$ elements, which are stored in a lookup table according to the power of each element.

**Example** **1.**
*To construct the field $GF\left(2^{3}\right),n=3$ a new element α is added to the field GF(2). α is a root of the primitive polynomial $P\left(x\right)=x^{3}+x+1$ with a degree of n=3, which is used to generate the elements of $GF\left(2^{3}\right)$. Since α is a root of the polynomial $P\left(\alpha \right)=\alpha^{3}+\alpha +1=0$ then, $\alpha^{3}=\alpha +1,\alpha^{4}=\alpha \alpha^{3}=\alpha \left(\alpha +1\right)$ and so on. The field $GF\left(2^{3}\right)$ has $2^{3}=8$ elements.*


In Table 3, a $\alpha =\alpha^{1}$ is introduced as a root, but $\alpha^{0}=1$ and 0 can also be introduced, since they are do have representation in the field. In the case of a $GF(2^{16})$ field containing $2^{16}=256$ elements, the same method can be utilised to organise the elements for use in 16 × 16 MIMO arrays.

### 4.3. AI-Based Procedures to Replace Human Processing

In Section 4.2, the interaction between Artificial Intelligence algorithms and modulation/demodulation was described. At both the transmitter and receiver, the Scikit-learn Python module is employed [75].

#### 4.3.1. Logistic Regression

The weighted sum of the input attributes is used in logistic and linear regressions. However, the logistic regression bias has a binary output as opposed to a direct output. According to Suykens [76], a logistic regression model predicts that if the probability is less than 50%, it belongs to the negative class denoted by “A” or “0”, and if it is greater, it belongs to the positive class denoted by “B” or “1”.

To find the value of the prediction, Equation (Equation 21) can be used:(21)$$\frac{\partial }{\partial \theta_{j}}MSE\left(\theta \right)=\frac{2}{m}\sum_{i=1}^{m}\left(\theta^{T}\cdot x^{\left(i\right)}-y^{\left(i\right)}\right)x_{j}^{\left(i\right)}\;,$$
where *m* is the number of partial derivatives, x is the input, and *y* is the predicted value. Equation (Equation 22) represents the logistic regression model’s estimated probability in vector form $\hat{p}$:(22)$$\hat{p}=h_{\theta }\left(x\right)=\sigma \left(\theta^{T}\cdot x\right)\;,$$
where θ is the vector of the model parameters, $\theta^{T}$ is the transpose of θ, $h_{\theta }$ is the hypothesis function, and σ(·), a logistic or logit sigmoidal function, generates a number between 0 and 1, as shown in Equation (Equation 23).
(23)$$\sigma =\frac{1}{1+exp(-t)}$$

After estimating the probability $\hat{p}=h_{\theta }\left(x\right)$ that an instance *x* belongs to the positive class, the Logistic Regression model can easily make its prediction $\hat{y}$. The logistic regression model’s prediction is shown in Equation (Equation 24).
(24)$$\hat{y}=\left\{\begin{array}{c} 0\;if\;\hat{p}<0.5\\ 1\;if\;\hat{p}\geq 0.5 \end{array}\right.$$

Note that σ(t)<0.5 when t<0, and σ≥0.5 when t≥0, so a logistic regression model predicts 1 if $\theta^{T}\cdot x$ is positive and 0 if is negative.

#### 4.3.2. Naive Bayesian with Gaussian optimisation

The Naive Bayesian with Gaussian optimisation (GaussianNB) method finds promising parameter values by using a Gaussian process model of the objective function [77]. The Probability of Improvement (PI) is an intuitive strategy that can be calculated analytically by using Gaussian processes to maximise the probability of improvement over the best current value [78]:(25)$$a_{PI}\left(x;\left\{x_{n},y_{n}\right\},\theta \right)=\Phi \left(\gamma \left(x\right)\right)\;,$$
(26)$$\gamma \left(x\right)=\frac{f\left(x_{best}\right)-\mu \left(x;\left\{x_{n},y_{n}\right\},\theta \right)}{\sigma (x;\left\{x_{n},y_{n}\right\},\theta )}\;,$$
where f:χ→R, $x_{best}$ is the best current value, μ is its predictive mean function and σ is the predictive variance function. In order to maximise the expected improvement over the best current value, the Expected Improvement (EI) could also be calculated using a Gaussian process:(27)$$a_{EI}\left(x;\left\{x_{n},y_{n}\right\},\theta \right)=\sigma \left(x;\left\{x_{n},y_{n}\right\},\theta \right)\left(\gamma \left(x\right)\Phi \left(\gamma \left(x\right)\right)+N\left(\gamma \left(x\right);0,1\right)\right)\;,$$
where $a_{EI}$ represents the acquisition function with the highest expected improvement, Φ(·) represents the cumulative distribution function and N represents the normal distribution. The upper confidence limit of Gaussian Processes (GP) seeks to exploit the concept of lower and upper confidence limits in the maximisation case in order to build acquisition functions that minimise regret as optimisation progresses [79]:(28)$$a_{LCB}\left(x;\left\{x_{n},y_{n}\right\},\theta \right)=\mu \left(x;\left\{x_{n},y_{n}\right\},\theta \right)-\kappa \sigma \left(x;\left\{x_{n},y_{n}\right\},\theta \right)\;,$$
where $a_{LCB}$: $\chi \to R^{+}$ denotes the acquisition function, LCB is the lower confidence bound and κ is tunable to balance exploitation versus exploration.

#### 4.3.3. SVM Classifier

A support vector machine divides the elements of a set into different subsets known as classes with the goal of finding the widest possible hyperplane that best separates these classes. The margin can be seen in Figure 5. It is defined as the maximum width of the region parallel to the hyperplane that has no interior data points. Equation (Equation 29) shows how a linear SVM predicts the class of a new *x* instance by calculating the decision function $w^{T}x+b=w_{1}x_{1}++w_{n}x_{n}+b$: if the result is positive, the predicted class f(x) is the positive class (1); otherwise, it is the negative class (0) [75]. *b* is the bias and *w* is the feature weight.
(29)$$f\left(x\right)=\left\{\begin{array}{ccc} 0 & \;\mathrm{si}\; & w^{T}x+b<0\\ 1 & \;\mathrm{si}\; & w^{T}x+b\geq 0 \end{array}\right.$$

To make it easier to separate the classes after this transformation, kernel functions move the data to a different, usually higher, dimensional space, potentially simplifying nonlinear complex decision boundaries in the assigned higher dimensional feature space to make them linear. The data do not have to be explicitly transformed in this process, which is known as a kernel trick [80]. A second-degree polynomial kernel is the function $K\left(a,b\right)={\left(a^{T}b\right)}^{2}$. Based on some mapping ϕ, the kernel *K* corresponds to an inner product in a feature space [81]. A kernel is a function in ML that computes the dot product $\phi {\left(a\right)}^{T}-\phi \left(b\right)$ by using only the original vectors a and b without computing the ϕ transformation. The polynomial kernel for polynomials of degree *d* is shown in Equation (Equation 30) [75].
(30)$$K\left(a,b\right)={\left(\gamma a^{T}b+r\right)}^{d}$$
where *a* and *b* are vectors in the input space, r≥0 is a free parameter that compensates for the impact of higher-order terms in the polynomial versus lower-order terms in the polynomial, and γ is a scaling parameter.

When r=0, it is said that the kernel is homogeneous. When d=1 and r=0 are implemented, the result is identical to that of a linear kernel. If *d* is greater than one, nonlinear decision limits are produced, with the degree of nonlinearity increasing as *d* increases. Due to overfitting, *d* values greater than 5 are typically not recommended. Figure 5 depicts the optimal hyperplane with a polynomial kernel separating the data, where the light blue and brown dots represent data belonging to two distinct classes. The segmented red lines represent the various hyperplanes that can be constructed to partition data representing two classes between two point clouds. In a similar fashion, the red line represents the hyperplane that maximises class separability.

#### 4.3.4. Extra Trees Classifier

The Extra Trees Classifier (ETC), also known as extremely random trees, generates a large number of decision trees, but the per-tree sampling is random. Tanha [82] used this method to assemble a data set with unique samples in each tree. According to Geurts [83], the geometric analysis generated by the ETC algorithm assumes a minimum number of samples $\left(n_{min}=2\right)$. When the number of trees is M→∞, the models generated by the Extra Trees algorithm appear to be linear. Thus, with the minimum sample condition $n_{min}\geq 2$, the algorithm can be extrapolated for the n-dimensional case. In this way, a continuous multilinear approximation is obtained for the case of infinite samples N→∞. In either case, the expression presented in Equation (Equation 31) can be used, where $x_{i}=\left(x_{1,i},\cdots ,x_{n,i}\right)$ is a *n* dimensional input vector, yielding $y_{i}$ as the output. To simplify the notation, we give the notation presented in the Equation (Equation 32), where *j*th indicates the value of the sample, so that $\forall \left(i_{1},\cdots ,i_{n}\right)\in {\left\{0,\cdots ,N\right\}}^{n}$. For $I_{\left(i_{1},\cdots ,i_{n}\right)}\left(x\right)$, the characteristic function of the hyperplane corresponds to the one presented in Equation (Equation 33).
(31)$$ls_{N}=\left\{\left(x_{i},y_{i}\right):i=1,\cdots ,N\right\}$$
(32)$$x_{j,(0)}=-\infty \;and\;x_{j,(N+1)}=+\infty ,\forall j=1,\cdots ,n$$
(33)$$\left[x_{1,i_{1}},x_{1,(i_{1}+1)}\right]\times \cdots \times \left[x_{n,(i_{n})},x_{n,(i_{n}+1)}\right]$$

This enables us to demonstrate that, as stated in Zhao [84], an infinite number of extra trees will generate an approximation of the form presented in Equation (Equation 34). Thus, for fully developed trees, the development shown in Equation (Equation 35) is shown.
(34)$$\hat{y}\left(x\right)=\sum_{i_{1}=0}^{N}\cdots \sum_{i_{n}=0}^{N}I_{\left(i_{1},\cdots ,i_{n}\right)\left(x\right)}\;\sum_{X\subset \{x_{1},\cdots ,x_{n}\}}\lambda_{\left(i_{1},\cdots ,i_{n}\right)}\prod_{x_{j}\in X}x_{j}$$
(35)$$\hat{y}\left(x_{i}\right)=y_{i},\forall \left(x_{i},y_{i}\right)\in ls$$

A piecewise linear model is obtained for the specific case of a one-dimensional input, as shown in Equation (Equation 36), where $I_{\left(i\right)}\left(x_{1}\right)$ is the interval characteristic function, and the values of $\lambda_{i,\phi }$ and $\lambda_{i,x_{1}}$ are obtained from Equations (Equation 35) and (Equation 36).
(36)$$\hat{y}\left(x\right)=\sum_{i_{1}=0}^{N}I_{\left(i_{1}\right)}\left(x\right)\sum_{X\subset \{x_{1}\}}\lambda_{(i_{1}),X}\prod_{x_{j}\in X}x_{j}=\sum_{i=0}^{N}I_{\left(i\right)}\left(x_{1}\right)\left(\lambda_{i,\phi }+\lambda_{j,s\{x_{1}\}}x_{1}\right)$$

#### 4.3.5. Histogram Gradient Boosting Classifier

Decision trees also inspired gradient boosting, one of the most useful algorithms for generating table structures and enabling predictive regression modelling, according to Padhi [85]. There are two variants that are based on the operating system implementation: Light Gradient Boosting (LGB) and GPU-accelerated XGBoost. LGB is a fast, distributed, high-performance gradient boosting framework based on the decision tree algorithm that can be used for ranking, classification and a variety of other ML tasks [86,87]. This model reduces the learning process time by at least 20 times while maintaining the same precision. According to Chen [88], the XGBoost algorithm boosts the GPU performance by using perfect shuffling of indexes and data in parallel sums and GPU-accelerated sorting, generating trees of all data concurrently for each iteration.

XGBoost is an enhanced version of the gradient boosting algorithm that is more efficient and scalable. Automatic feature extraction is one of the characteristics that distinguishes XGBoost from other algorithms. XGBoost supports regularisation to prevent overfitting and has the capacity to learn from nonlinear datasets. In addition, the parallelisation feature enables XGBoost to utilise multiple CPU cores. It is one of the tree-based additive ensemble models that consists of a group of base learners. XGBoost can generally be represented by:(37)$$F=\left(m_{1},m_{2},m_{3},m_{4}\ldots m_{n}\right),{\hat{y}}_{i}=\sum_{t=1}^{n}m_{t}\left(x_{i}\right)$$
where ${\hat{y}}_{i}$ is the final predictive model, which is the combination of all weak learners, and x is the input feature for each weak learner, i.e., m.

From the paper [87], we extracted the objective function for XGBoost, as given below:(38)$$\mathrm{Obj}\left(\theta \right)=\sum_{i=1}^{m}L\left(z_{i},z_{i}\right)+\sum_{t=1}^{T}\Omega \left(f_{t}\right).$$

In Equation (Equation 38), note that the objective function has two parts; the first part denotes the loss function, i.e., *L* denotes the training loss of either the logistic or squared loss, and the second part represents the addition of each tree’s complexity. $z_{i}$ is the actual value and $z_{i}$ is the predicted value, whereas Ω is the regularisation term, *T* denotes the total number of trees, and f is the function.

#### 4.3.6. Model Evaluation

Figure 6 depicts a classification table displaying the various error types.

The formula for calculating the recall parameter, which is relevant for the assessment of type 2 errors or false negatives, is presented in Equation (Equation 39):(39)$$recall=\frac{TP}{TP+FN}\;,$$
where TP stands for True Positives and FN stands for False Negatives [89].

## 5. Results Analysis

The interface illustrated in Figure 7 corresponds to Galois Field Mapping/Galois Fields Demapping. During the phase of finding the ideal algorithm, the transmitter converts images to polynomials (shown from left to right), while the receiver converts polynomials to images (shown from right to left), as displayed in Figure 7. The time has come to reveal the findings. First, the results of the channel based on CSK/QAM mapping will be shown, followed by the SARS-CoV-2 Searching results of the model, and finally, the SARS-CoV-2 Operation results of the best model.

### 5.1. Results of the Channel Based on CSK/QAM Mapping

The communications channel is the source of all negative effects when collecting data for the purpose of locating and implementing the optimal model. Traditionally, IEEE 802.15.7 specifies rules for designing M-CSK constellations and directly applying them for modulation. However, the mapping defined in Section 4.1 can also be used to indicate which cursor to map. Then, use a square constellation with better separation properties can be used. Figure 8 and Figure 9 illustrate the experimental results for $N=2^{2^{i}}\times 2^{2^{i}},i=0,1$. Despite their two-dimensional representation in the CIE xyY plane, the null element points can be aggregated with an integer cursor can be assigned to each of them, and a table can be created in the cloud to provide copies at the transmitter and receiver for calculating inverse mapping between the M-CSK and the M-QAM constellation.

Figure 10 shows the MIMO channel capacity for $N=2^{2^{i}}\times 2^{2^{i}}\left(i=0,1,2,3\right)$. In order to get a greater spectrum efficiency for a longer chromosome, it has been demonstrated that square constellations should be favoured due to their superior separation properties, despite their exponentially increasing complexity.

It is feasible to transmit monochromatic photon streams. When coherent monochromatic frequencies are employed, energy is not wasted on phase incoherence effects, which ordinarily result in self-destructive phase effects. Consequently, the outcomes are improved. Due to the fact that quantum objects do not require their attributes to be specified, a beam of baseband light can arrive at the photodetector from a variety of angles other than a coherent straight line. The use of a laser decreases costs because white light amplification equipment is avoided due to the high concentration of energy in a single frequency tone.

In order to generalise the FSK channel to a MIMO channel [51], Figure 11 and Figure 12 show the output of the MIMO demodulator that will enter the “Galois Field demapping” process for 256-point QAM square constellation with SNR values of SNR=60[dB] and SNR=30[dB].

Figure 13 compares CSK/QAM mapping to QAM mapping by using the XGBoost algorithm in terms of the BER for different SNR values with $N=2^{2^{i}}\times 2^{2^{i}},i=0,1,2,3$. This simulation was carried out in 1[dB] steps for values up to SNR=80[dB]. The BER was computed within one cycle using the MMSE estimator.This procedure was repeated 10,000 times to accumulate the erroneous values for each of the SNR[dB] levels in an array. In contrast to linear mapping, the combination of M-CSK and M-QAM mapping results in nonlinear productions, i.e., it breaks the regularity of selecting the same points due to the centroid calculation and the inclusion of points as the null element. As mentioned in Section 4.3.5, the XGBoost algorithm produces the best results because it manages to generate multiple trees and is the only one capable of learning a nonlinear dataset, because it generates a new objective function. When the dataset is small, it cannot learn to predict the values to come, for example, when $N=2^{2^{i}}\times 2^{2^{i}},andi=0,1$ points occur, although it is seen to start working for $N=2^{2^{i}},i=3$, but when the dataset increases, the gain is significantly improved. For BER=$10^{-3}$, gains of −10 [dB], −3 [dB], 3 [dB] and 5 [dB] occur for $N=2^{2^{i}}\times 2^{2^{i}},i=0,1,2,3$, respectively. It is concluded that the square constellation $N=2^{2^{i}}\times 2^{2^{i}},i=3$ produces a greater benefit. When the data set is small, the algorithm is unable to learn to forecast future values. Alternatively, the BER improves as the data set is enlarged, making it simpler to separate states and creating a larger forest.

### 5.2. Results SARS-CoV-2 Searching of the Model

Based on Section 4, this subsection analyses the proposed strategies in order to gather the information required for model searching and the operation of the best model from Section 5.3. The obtained biological material is subjected to the thermocycling process depicted in Figure 1, and the results are subsequently deposited on the electrophoresis gel, which is imaged after the reaction occurs. This is illustrated in Figure 14.

In addition to the channel noise described previously, the sample includes a significant amount of background noise. The use of a denoising convolutional auto-encoder model contributes to enhancement of the sample quality [32]. Figure 15 presents a comparison of the input and output images, with the output image containing less background noise. The detection of bands is the second step in the image processing procedure. Figure 16 depicts the outcomes of applying the methodology. The bands can be segmented with the data that will be subsequently analysed.

Figure 17 depicts data obtained from one of the bands. The top image shows the automatic clipping of the ladder band and the bottom image shows the average curve. Figure 18 show the recognition of peaks in the average curve obtained from the bottom image in Figure 17. This allows numerical representation through the method discussed in Section 4.2. Figure 16 and Figure 17 show the average bands $P_{x}^{AVG}$ and $P_{y}^{AVG}$. This information is used to train various mathematical models that enable band classification from electrophoresis bands. Figure 19 depicts the outcome of applying the Pearson correlation between the bands, demonstrating how the correlation of “band0” to the other bands is too low in comparison with the other values, which are higher than 0.5%. This produces an accuracy of 100%. Unfortunately, it cannot be used for the classification of positive and negative samples due to the high level of error.

Based on these data, classification models can be trained to differentiate between three categories: ”Positive (+),” ”Negative (−)” and “Ladder”. The training results of the models described in Section 4.3 are shown in Figure 20. In Table 4, it can be seen that the XGBoots classifier has the highest training accuracy of 96.03% compared with the other models and a recall rate of 99% for positive values.

### 5.3. Results SARS-CoV-2 Operation of the Best Model

ML involves the parallel calculation of all processes and the selection of the best one; however, the processes may be conducted sequentially depending on the computer available. Images may be presented to Galois Fields mapping or switched directly to CSK/QAM modulation, depending on how the operation is configured, by entering the cursor *k* from Equation (Equation 41).

In this instance, the most important parameter to investigate is recall, which indicates how frequently the model generates type 2 errors. In terms of both this metric and precision, the XGBoost classifier model has the best performance. The associated parameters are displayed in Table 4. It can be seen that the model classifies the ladder correctly in all instances. It is important to note that recall is computed using Equation (Equation 39). Figure 14 depicts the results of the application of this method to the image shown in Figure 21. This corresponds to the first three boxes in the transmitter and the last three boxes in the receiver in Figure 1, which were previously completed manually but are now performed by an ML subsystem.

The accuracy of trained models will always be determined by the criteria used by the medical professional who prepared the data set. Due to uncertainty, the data can be propagated to the VLC channel. At the time of operation, the developed system is merely a tool; the results must be confirmed and interpreted by another health professional to determine the presence or absence of SARS-CoV-2. The created system reduces sample recognition times, allowing professionals to make more accurate diagnoses, and expands the data set size.

## 6. Conclusions

In this research work, an innovative VLC-based method for detecting COVID-19 in a subterranean environment was proposed. It was found that the unfavourable effects of the underground channel on VLC communications can be mitigated through precise mathematical modelling of the underground channel.

In order to get a higher spectrum efficiency for longer chromosomes, it has been shown that square constellations should be favoured due to their superior separation qualities within a photon stream. Transmission of monochromatic photon streams is an additional alternative. Since no energy is expended on phase incoherence effects, which generally result in self-destructive phase effects, when coherent monochromatic frequencies are employed, the results improve. It was also established that employing a laser saves money because there is no need for white light amplification equipment, often known as a driver. In addition, it was revealed that mathematical scaffolding in the exponential representation of DNA, in conjunction with the novel modulation and suitable channel modelling, prevents the transmission of heavy images.

The XGBoost technique was found to be the most successful, since it generates a large number of trees and is the only one that can learn a nonlinear data set by creating a novel goal function. When $N=2^{2^{i}}\times 2^{2^{i}},i=0,1$ points are used. For example, the dataset is too small for the algorithm to learn to predict future values, despite a slight improvement when $N=2^{2^{i}}\times 2^{2^{i}},i=1$ points are used. As the dataset expands in size, the gain increases dramatically; this is something that linear models cannot achieve. For BER = $10^{-3}$, gains of −10 [dB], −3 [dB], 3 [dB] and 5 [dB] were achieved for $N=2^{2^{i}}\times 2^{2^{i}},i=0,1,2,3$, respectively. The conclusion is that the square constellation $N=2^{2^{i}}\times 2^{2^{i}},i=3$ yields a greater profit. During the searching phase, a classification algorithm was selected from a pool of available options. For a total of 630 COVID-19 samples, the best model was XGBoots, which displayed an accuracy of 96.03% and a recall rate of 99% for positive values, placing its performance above that of the other models.

Furthermore, the uncertainty in the data propagates to the channel, so the accuracy of the trained models is determined by the criteria employed by the expert who creates the dataset. Clearly, the only way to rectify this is to compile a dataset from multiple sources so that it is complete and objective.

By extracting genetic information more efficiently, it is possible to classify the bands present in electrophoresis samples by using ML and a three-state classification process to determine whether the band corresponds to a COVID-19 positive, negative or ladder sample.

## Figures and Tables

**Figure 1 sensors-23-01533-f001:**
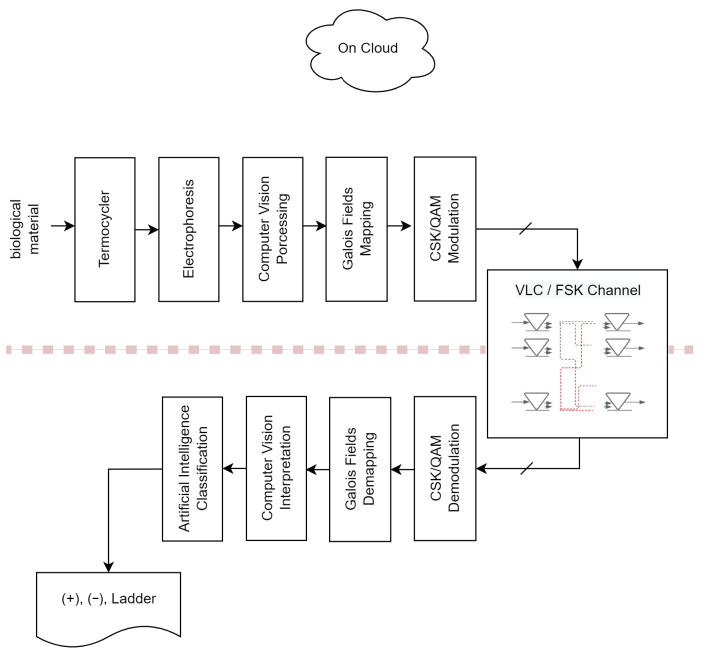
System diagram for the phases of model searching and operation of the classification model.

**Figure 2 sensors-23-01533-f002:**
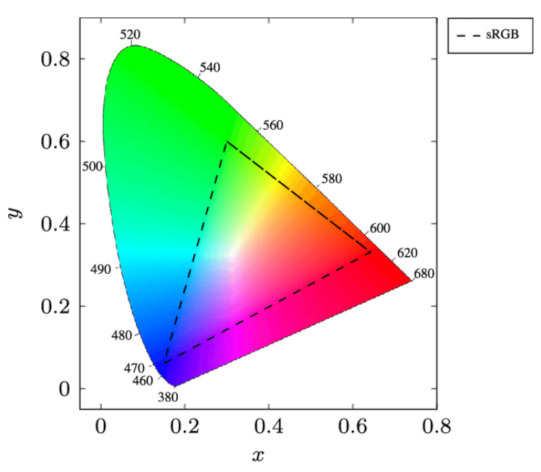
The xyY diagram of the Commission Internationale de l’Éclairage.

**Figure 3 sensors-23-01533-f003:**
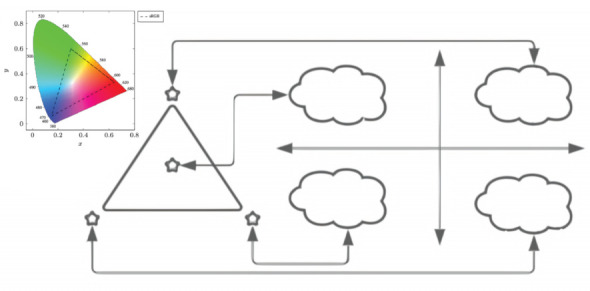
General CSK/QAM assignment.

**Figure 4 sensors-23-01533-f004:**
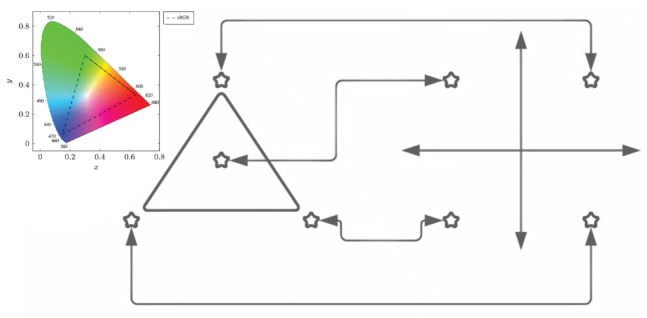
4-CSK constellation mapping into 4-QAM.

**Figure 5 sensors-23-01533-f005:**
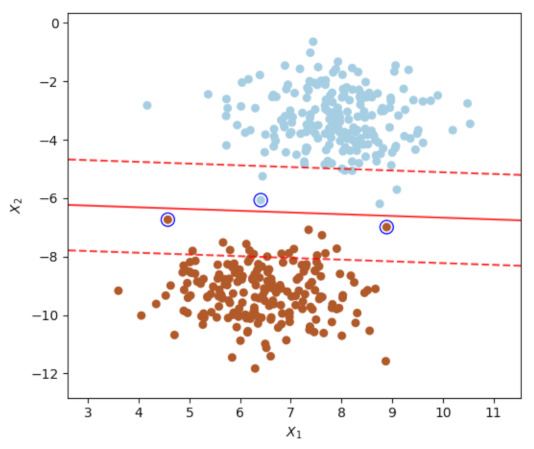
Graph of a polynomial SVM showing the hyperplane separating the samples from the classes.

**Figure 6 sensors-23-01533-f006:**
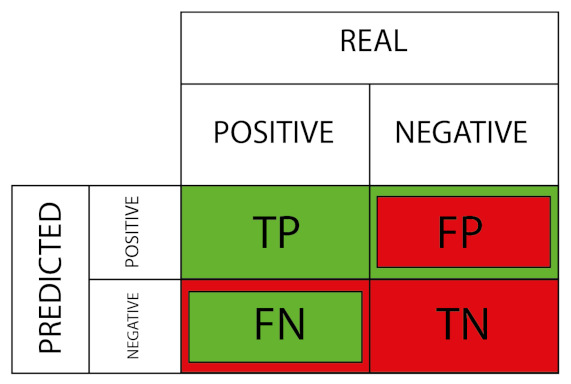
Outputs of the classification model.

**Figure 7 sensors-23-01533-f007:**
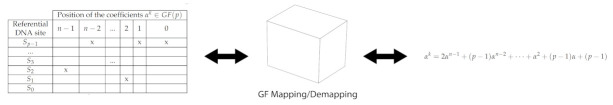
Galois Field Mapping/Galois Fields Demapping procedure.

**Figure 8 sensors-23-01533-f008:**
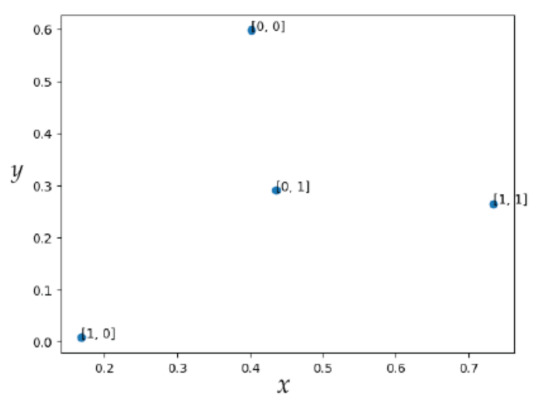
CSK constellation for $N=2^{2^{i}}\times 2^{2^{i}},i=0$.

**Figure 9 sensors-23-01533-f009:**
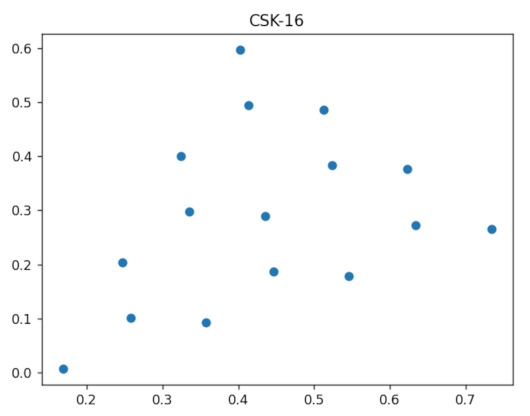
CSK constellation for $N=2^{2^{i}}\times 2^{2^{i}},i=1$.

**Figure 10 sensors-23-01533-f010:**
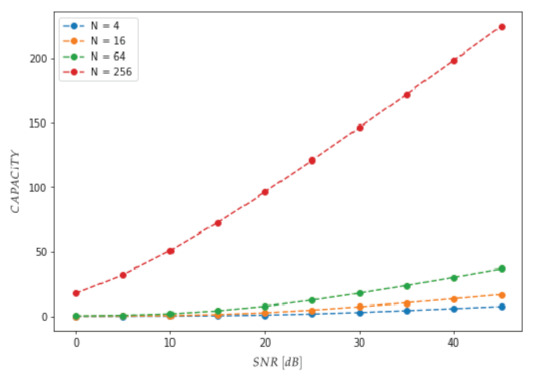
MIMO channel capacity for $N=2^{2^{i}}\times 2^{2^{i}}\left(i=0,1,2,3\right)$.

**Figure 11 sensors-23-01533-f011:**
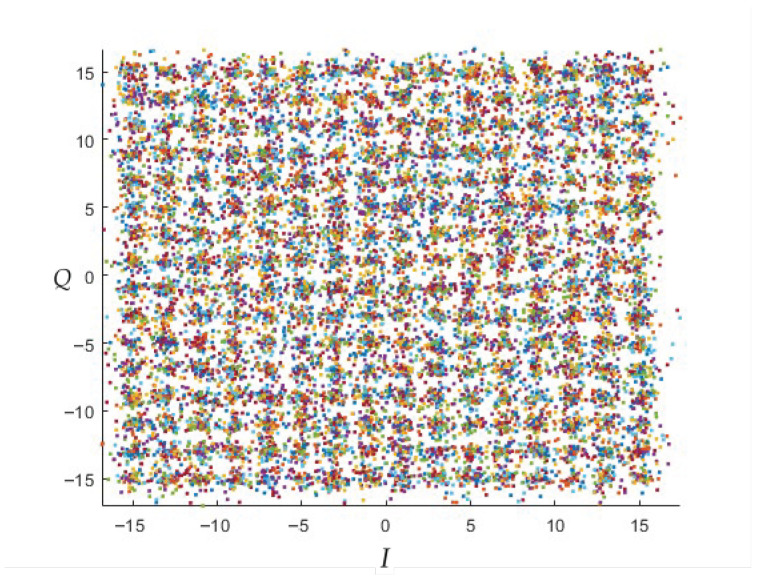
QAM constellation for $N=2^{2^{i}}\times 2^{2^{i}},i=3$ on the MIMO channel, SNR=60[dB].

**Figure 12 sensors-23-01533-f012:**
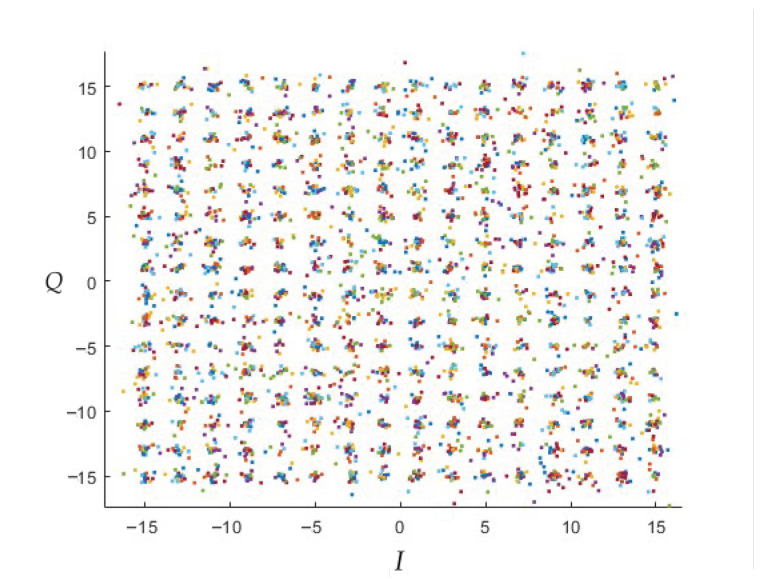
QAM constellation for $N=2^{2^{i}}\times 2^{2^{i}},i=3$ on the MIMO channel, SNR=30[dB].

**Figure 13 sensors-23-01533-f013:**
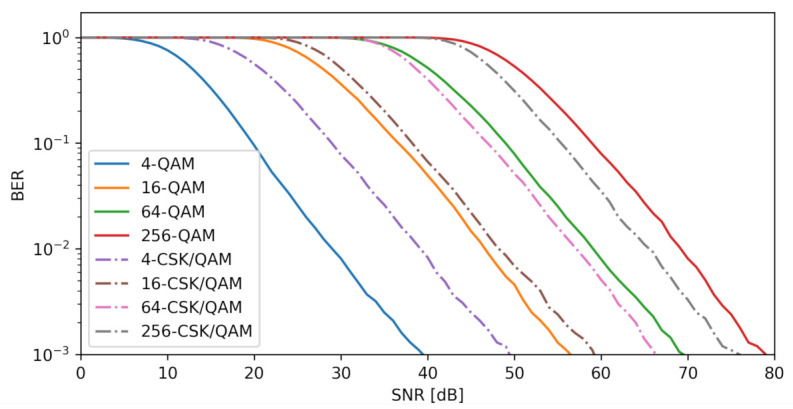
BER at different SNR levels in a $N=2^{2^{i}}\times 2^{2^{i}}$ MIMO array for i=0,1,2,3.

**Figure 14 sensors-23-01533-f014:**
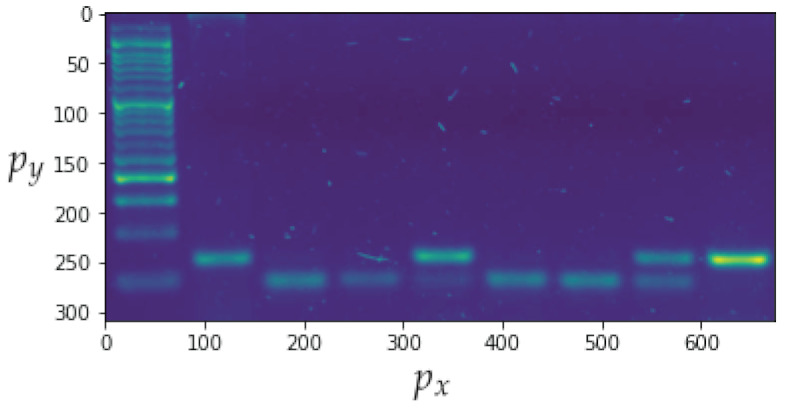
Image of the electrophoresis gel where $p_{x}$ is the position of the horizontal pixel and $p_{y}$ the position of the vertical pixel.

**Figure 15 sensors-23-01533-f015:**
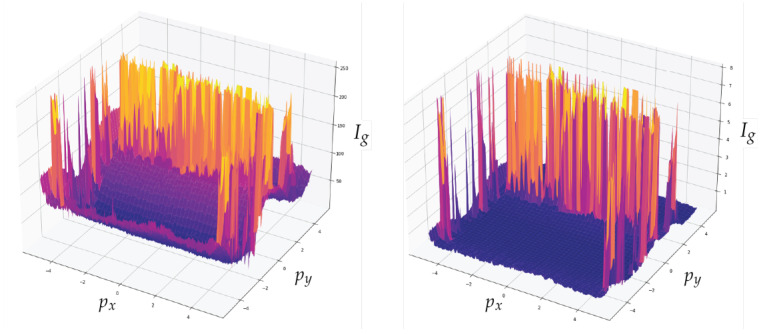
Comparison between the input image and the output of the convolutional denoising autoencoder model, where $p_{x}$ is the position of the horizontal pixel, $p_{y}$ is the position of the vertical pixel and $I_{g}$ is the grayscale intensity of the pixel.

**Figure 16 sensors-23-01533-f016:**
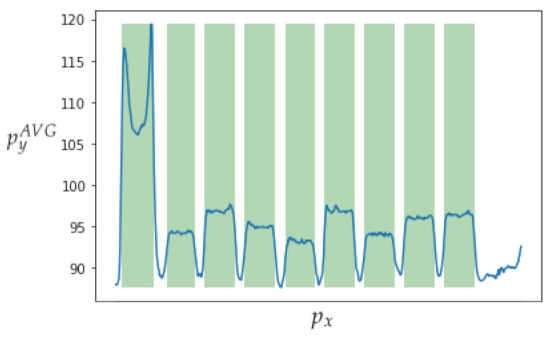
Detection of bands in an electrophoresis sample where $p_{y}^{AVG}$ is the average value of $p_{y}$ and $p_{x}$ is the horizontal value of the pixel.

**Figure 17 sensors-23-01533-f017:**
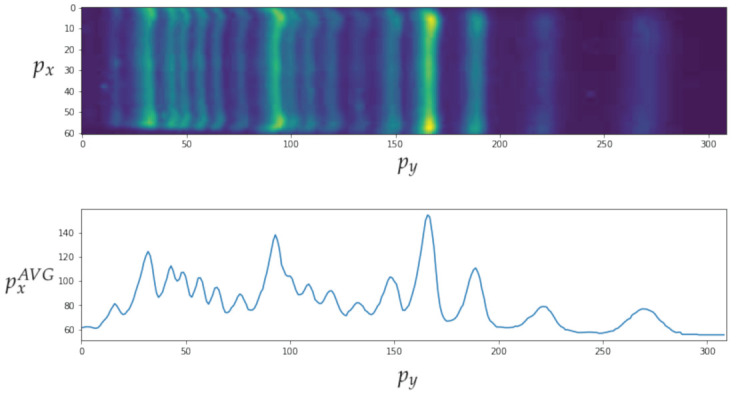
The top image shows the automatic clipping of the ladder band. The bottom image shows the average curve, where $p_{y}$ is the vertical position and $p_{x}^{AVG}$ is the average horizontal value $p_{x}$.

**Figure 18 sensors-23-01533-f018:**
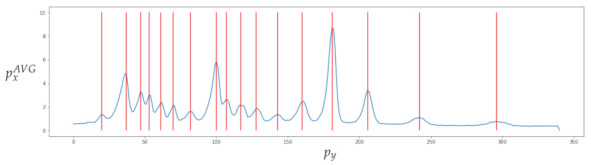
Recognition of peaks in the average curve obtained from the bottom image in Figure 17.

**Figure 19 sensors-23-01533-f019:**
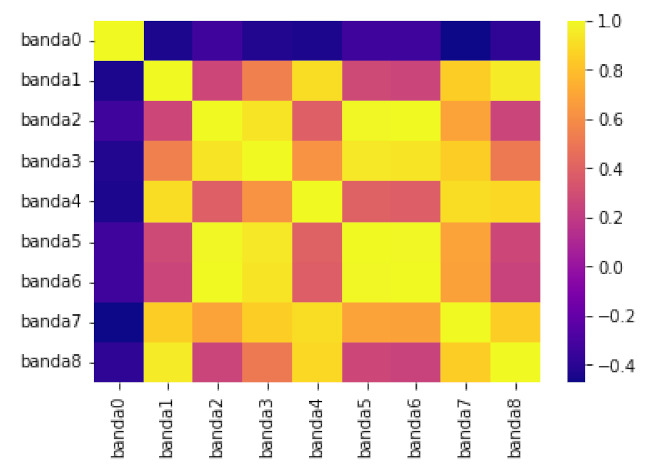
Application of Pearson’s correlation on the different columns of the electrophoresis sample.

**Figure 20 sensors-23-01533-f020:**
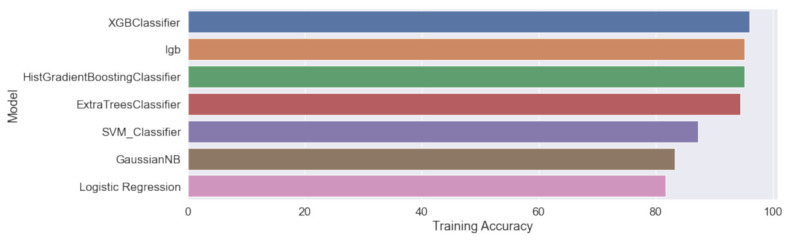
Comparison of the accuracy levels of the different trained classification models.

**Figure 21 sensors-23-01533-f021:**
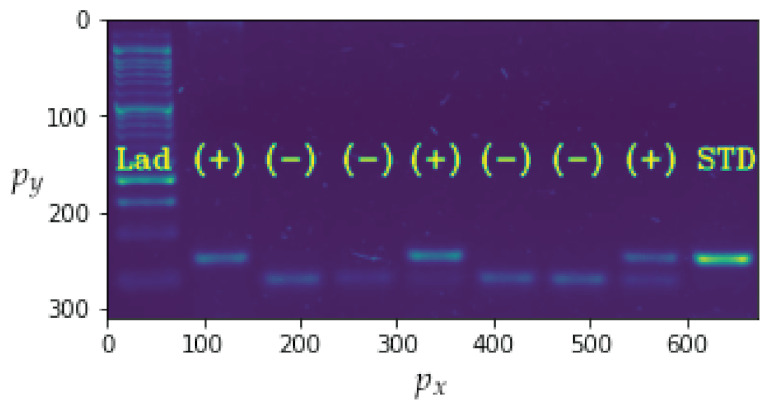
Results of the application of XGBoots.

**Table 1 sensors-23-01533-t001:** DNA representation for elements in GF(p).

r∈GF(p)	0	1	2	⋯	p−1
Size of DNA Fragment [bp]	$S_{0}$	$S_{1}$	$S_{2}$		$S_{p-1}$

**Table 2 sensors-23-01533-t002:** dsDNA fragment representation of $\alpha^{k}$ performed by agarose gel electrophoresis.

	Positions of the Coefficients $\alpha^{k}\in GF\left(p\right)$					
**Referential DNA Site**	n−1	n−2	…	2	1	0
$S_{p-1}$		x			x	x
*…*						
$S_{3}$			…			
$S_{2}$	x					
$S_{1}$				x		
$S_{0}$						

**Table 3 sensors-23-01533-t003:** Generation of the polynomials from $P\left(x\right)=x^{3}+x+1$.

Element $GF\left(2^{3}\right)$	Polynomial	Symbol
0	0	000
$\alpha^{0}$	1	001
$\alpha^{1}$	α	010
$\alpha^{2}$	$\alpha^{2}$	100
$\alpha^{3}$	α+1	011
$\alpha^{4}$	$\alpha^{2}+\alpha$	110
$\alpha^{5}$	$\alpha^{2}+\alpha +1$	111
$\alpha^{6}$	$\alpha^{2}+1$	101

**Table 4 sensors-23-01533-t004:** Parameters obtained from the training of the XGBoost classifier.

**Training Accuracy**	100.0%			
**Model Accuracy Score**	96.03%			
**Classification Report**				
	**precision**	**recall**	**f1-score**	**support**
Ladder	1.00	1.00	1.00	17
Positive (+)	0.95	0.99	0.97	85
Negative (−)	0.95	0.83	0.89	24
Accuracy			0.96	126
Macro avg	0.97	0.94	0.95	126
Weighted avg	0.96	0.96	0.96	126

## Data Availability

Not applicable.

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
