# Peer review of "A New COVID-19 Detection Method Based on CSK/QAM Visible Light Communication and Machine Learning"

_sensors, 2023, doi:10.3390/s23031533_

Round 1

Reviewer 1 Report

This paper proposed Covid-19 detection in an underground tunnel. The article is very confusing and there is no flow in the sense that the authors want to convey.

1. For example, the title itself is so long and uses VLC and RF and CSK/QAM both. I can't understand the title itself. The authors should simplify their title so that the common reader can understand. 

2. The structural write-up is a bit poor and the technical description at some points is very vague. The authors should have used a grammar check editor for some grammatical errors and misprints. 

3. It is unclear how 6G is used in Covid-19 detection. 

4. The significance of the study is very less now as Covid-19 is already a curable disease now. Please highlight the motivation and significance of the study. 

Overall, the paper is too long, and confusing, without any structure and motivation. I cannot recommend its publication in its current form. 

Reviewer 2 Report

This paper《 PROPOSED COVID-19 DETECTION IN AN UNDERGROUND TUNNEL USING HYBRID VLC/RF AND CSK/QAM MAPPING FOR FUTURE 6G》presents a system for detecting Coronavirus disease 2019 (COVID-19) inside an underground tunnel using the sixth generation of mobile devices (6G), which has a clear significance and theoretical reference value on how to continuously monitor pathogens in underground tunnels. However, the following problems still exist:

1.  The second section is only the listing of the literature, the content of the lack of logic and coherence, please re-comb and summarize the status quo of research at home and abroad.

2.  Using the full name when the noun first appears. For example, LOS, NLOS in 2.4.2.

3.     Section 3 doesn’t explain why some of the samples were used for training and what the benefits were.

4.     The Kuhn-Tucker (KT) conditions and the dual space in 4.2.3 are not explained.

5.     The 4-CSK constellation diagram of 4-CSK in 4.4.4 is poorly described and the parameters are represented on the diagram as much as possible, and it is recommended to distinguish it from the CIE 1931 color space chromaticity diagram;

6.     In 4.5, the channel analysis is not in place and the specific parameters are not given; meanwhile, the parameters in figure 4,5 are not explained.

7.     Please check if BFSK is in figure 21 in detail.

8.     When describing the mapping in Figure 8, you do not explain the reasons for its adoption.

9.  The paper does not highlight the work done by the author.

To sum up,the paper proposes a system for the sustainable monitoring of pathogens in underground tunnels at 6G. The idea is novel and has definite significance for how to continuously monitor the pathogen in the underground tunnel. However, there are still some problems in the article, such as illogical statements, repetitive statements, unclear statements, and lack of emphasis on research. Therefore, I suggest that the author compress the content, highlight the core content, reflect the work done. It is suggested to continue reviewing the manuscript after revision.

Round 2

Reviewer 1 Report

The authors addressed most of the comments. There are a few minor comments:

1. The title should be in capitalize case and not uppercase. 

2. In optical camera communications, the authors should cite state-of-the-art research. They can cite the following papers:

--------, "A survey of design and implementation for optical camera communication," Signal Processing: Image Communication, Volume 53, 2017, Pages 95-109. https://doi.org/10.1016/j.image.2017.02.001 

--------------, "Vehicle positioning based on optical camera communication in V2I environments," Computers, Materials & Continua, vol. 72, no.2, pp. 2927–2945, 2022. https://doi.org/10.32604/cmc.2022.024180 

3. The structure of the paper is still irregular. I will suggest the author merge several small paragraphs to make big paragraphs. There are several small paragraphs in the paper which makes the look of paper bad. 

Author Response

Santiago December 11, 2022

Cover Letter

Dear Editor

Thank you very much for your mail. We attach second correction of the manuscript titled " PROPOSED COVID-19 DETECTION IN AN UNDERGROUND TUNNEL USING HYBRID VLC/RF AND CSK/QAM MAPPING FOR FUTURE 6G".

Based on your suggestions, we have used the linked version and checked all references for relevance.

Please note that in the second round, Reviewer 1 specified that: "English language and styling is fine or minor fixes are needed", so we have made minor fixes, which are described in the "Response to Reviewer 1 Comments (Second Round)" file. Additionally, the same file contains one correction for the first and second comment, while thirty corrections for the third. All of which are highlighted in yellow.

Best Regards

Ismael Soto
